# Clinical Impact of CDK4/6 Inhibitors in De Novo or PR− or Very Elderly Post-Menopausal ER+/HER2− Advanced Breast Cancers

**DOI:** 10.3390/cancers15215164

**Published:** 2023-10-26

**Authors:** Hiu Tang, Daniel Yeo, Karen De Souza, Omar Ahmad, Tahir Shafiq, Okezie Ofor, Anjana Anand, Syed Karim, Sarah Khan, Srinivasan Madhusudan

**Affiliations:** 1Department of Oncology, Nottingham University Hospitals NHS Trust, Nottingham NG5 1PB, UKsarah.khan@nuh.nhs.uk (S.K.); 2Department of Oncology, University College London Hospitals NHS Foundation Trust, London NW1 2PG, UK; 3Biodiscovery Institute, School of Medicine, University of Nottingham, University Park, Nottingham NG7 3RD, UK

**Keywords:** advanced breast cancer, ER+/HER2− breast cancers, PR, CDK4/6 inhibitors, palbociclib, ribociclib, abemaciclib, PFS, OS

## Abstract

**Simple Summary:**

The cyclin-dependent kinase 4/6 (CDK 4/6) inhibitors palbociclib, ribociclib and abemaciclib have transformed the lives of patients with ER+/HER2− metastatic breast cancer (MBC). Clinical trials have shown that all three CDK4/6 inhibitors improve progression-free survival (PFS), but only ribociclib and abemaciclib improve overall survival (OS). The data have generated debate in the oncology community as to which CDK4/6 inhibitor to use in routine clinical practice. In this context, real-world data in a heterogeneous population of advanced breast cancer patients can inform clinical decision-making. Here, in a cohort of 227 patients, we show that palbociclib and ribociclib have similar PFS and OS benefits in the real world. Our data should reassure oncologists that both palbociclib and ribociclib remain effective treatment options for patients.

**Abstract:**

The CDK4/6 inhibitors significantly increase progression-free survival (PFS) in ER+/HER2− advanced breast cancer patients. In clinical trials, overall survival (OS) improvement has been demonstrated for ribociclib and abemaciclib but not for palbociclib. We undertook a real-world evaluation of PFS and OS in 227 post-menopausal patients who received first-line CDK4/6 inhibitors. There is no significant difference in median PFS (27.5 months vs. 25.7 months, *p* = 0.3) or median OS (49.5 months vs. 50.2 months, *p* = 0.67) in patients who received either palbociclib or ribociclib, respectively. De novo disease is significantly associated with prolonged median PFS and median OS compared with recurrence disease (47.1 months vs. 20.3 months (*p* = 0.0002) and 77.4 months vs. 37.3 months (*p* = 0.0003), respectively). PR– tumours have significantly reduced median PFS and OS compared with PR+ disease (19.2 months vs. 38 months (*p* = 0.003) and 34.3 months vs. 62.6 months (*p* = 0.02), respectively). In the very elderly (>80 years), median PFS and OS are significantly shorter compared with patients who are 65 years or below (14.5 months vs. 30.2 months (*p* = 0.01), and 77.4 months vs. 29.6 months (*p* = 0.009), respectively) in the palbociclib group. Our data suggest that the benefit in the very elderly is limited, and PR+/de novo disease obtains the maximum survival benefit.

## 1. Introduction

Cyclin-dependent kinases (CDKs) are a network of related serine/threonine kinases that, along with cyclins, control the cell cycle progression [1]. CDK4 and CDK6 drive progression through the G1/S phase of the cell cycle via retinoblastoma (Rb) protein signalling [2,3,4,5]. The CDK4-cyclin D1 complex translocates to the nucleus and phosphorylates the Rb protein, leading to de-repression of the E2F transcription factor, thereby promoting cell cycle progression and cellular proliferation [4,5,6]. In breast cancer (BC), CDK-cyclin pathways are constitutively activated by events, such as cyclin D1 overexpression [7,8]. Whilst nuclear CDK4 or CDK6 are involved in cell cycle regulation, there is also increasing evidence for a cytoplasmic function of CDK4/6. Cytoplasmic CDK4-cyclin D1 complexes may promote cancer cell migration and invasion and increase the metastatic potential [9]. In addition to its role in G1/S cell cycle regulation, CDK6 is also involved in transcriptional regulation of vascular endothelial growth factor A (VEGFA) (critical for angiogenesis) [10] or FMS-like tyrosine kinase 3 (FLT3) (involved in proliferation) [11]. CDK6 is positively correlated with angiogenesis in triple-negative breast cancer (TNBC) [12]. In addition, CDK6 kinase activity during oncogenic stress may promote cancerous transformation by antagonizing the p53 response in cells [13,14].

The highly selective CDK4/6 inhibitors, ribociclib, palbociclib and abemaciclib, target the ATP-binding domains of CDK 4 and 6. The CDK4/6 blockade induces G1 cell cycle arrest and cytostasis in an Rb-dependent manner [15,16]. Recent reports also indicate that CDK4/6i could indirectly inhibit CDK2 by preventing the formation of stable cyclin D-CDK4/6-p21/p27 complexes, thereby allowing p21 to block CDK2 activity. In addition, CDK4/6i can also induce a “senescence-like state”, promote epigenetic remodelling and autophagy, blockade oncogenic signalling networks and promote tumour immunogenicity [15,16].

Ribociclib, palbociclib and abemaciclib have transformed the lives of patients with ER+/HER2− metastatic breast cancer (MBC). In post-menopausal women, in the first-line setting, all three CDK4/6 inhibitors have shown significant improvements in progression-free survival (PFS) [17,18,19,20,21]. Overall survival (OS) benefit was demonstrated only in clinical trials of ribociclib [22] and abemaciclib [23] but not for palbociclib [24]. The OS differences seen in clinical trials have generated considerable debate around the optimal first-line choice in the metastatic setting. Differences in trial designs, missing survival data issues (such as loss of follow-up and withdrawal of consent) and potentially distinct biological activities of various CDK4/6 inhibitors have all been postulated to account for the observed differences in OS. However, real-world data (RWD) have the potential to clarify these ongoing debates and inform optimal management in routine clinical practice. Moreover, RWD can also inform the role of CDK4/6i in the very elderly (who are not usually included in clinical trials) and those with progesterone receptor (PR)– or de novo disease. We undertook a large single-centre retrospective study of patients who received CDK4/6i in the post-menopausal setting.

## 2. Materials and Methods

### 2.1. Patients

This retrospective clinical audit study was undertaken at Nottingham University Hospitals with approval from the service evaluation and quality improvement department (audit approval ID: 21-622C) for data presentation and publication. All post-menopausal patients with ER+/HER2− advanced breast cancer who received first aromatase inhibitor (anastrozole or letrozole or exemestane) endocrine therapy with a CDK4/6 inhibitor (palbociclib or abemaciclib or ribociclib) were included in the audit analysis. All patients provided informed consent for treatment. The patient demographics are summarized in Table 1.

### 2.2. Data Collection

The clinical team identified 227 ER+/HER2− post-menopausal MBC patients, who were treated with palbociclib, ribociclib or abemaciclib as per physician choice alongside an aromatase inhibitor (letrozole or anastrazole or exemestane) between 2017 and 2020. Baseline demographics were captured. Patients underwent regular clinical, biochemical and radiological surveillance with three monthly CT cross-sectional imaging as per protocol [17,18,19,20,21].

### 2.3. Histology

Expression of HER2, ER and PR was assessed according to the American Society of Clinical Oncology/College of American Pathologists (ASCO/CAP) guidelines [25,26]. For ER status, the EP1 clone was used (Dako-Cytomation, Santa Clara, CA, USA). For PR status, the PgR636 clone was used (Dako-Cytomation, Santa Clara, CA, USA). ER and PR assays were considered negative if there were <1% positive tumour nuclei in the presence of the expected reactivity of internal and external controls. For HER2 status, rabbit polyclonocal was used (Daka-Cytomation, Santa Clara, CA, USA). The HER2 test result was considered positive if the IHC 3+ membrane was positive or IHC2+/FISH+. The HER2 test result was considered negative if the IHC 0 membrane was negative. The HER2 test result was considered low if IHC1+ or IHC2+/FISH−. The histopathological characteristics are summarized in Table 2.

### 2.4. Treatment

All patients received CDK4/6i as per NICE recommendation and defined by the Cancer Drugs Fund (CDF), England. All patients had histologically confirmed ER+ and HER2− breast cancer. Patients had no previous hormone therapy for locally advanced or metastatic disease, i.e., endocrine therapy naïve for locally advanced/metastatic breast cancer. Previous hormone therapy with anastrazole or letrozole in the adjuvant or neoadjuvant setting was allowed, provided the patient had a disease-free interval of 12 months or more since completing treatment with neoadjuvant or adjuvant anastrazole or letrozole. All patients had an ECOG performance status of 0, 1 or 2. Palbociclib, ribociclib or abemaciclib therapy and dose modifications were as per standard protocols. Patients underwent regular clinical, biochemical and radiological surveillance during treatment including cross-sectional imaging.

### 2.5. Statistical Analysis

Overall survival (OS) and progression-free survival (PFS) rates were determined using the Kaplan–Meier curves. *p*-values of less than 0.05 were considered statistically significant. Hazard ratios and 95% confidence intervals were calculated using the Cox proportional hazard model. Survival curves were compared using the log-rank test among different groups of patients. All analyses were conducted using the statistical software GraphPad PRISM 9 for Mac (Prism 9, version 9.4.1, San Diego, CA, USA). PFS was defined as the time from initiation of CDK4/6 therapy to radiological progression, clinical progression or death. OS was defined as the time from diagnosis of metastatic disease to death from any cause. Data were censored on the 1st of March 2023 with a median follow-up of 49.5 months.

## 3. Results

### 3.1. Demographics and Characteristics

Patient demographics are summarized in Table 1. A total of 227 post-menopausal patients were suitable for PFS and OS analysis. In the whole cohort, the median age was 69 years. Overall, 57% of patients were aged 65 or older, with 35 (15.0%) patients aged 80 or above. The majority of the patient cohort was white Caucasian women. Overall, 57.0% of patients had recurrent disease, and 43.0% had de novo disease. More than half of the breast tumours were of invasive ductal subtype (56.8%). The invasive lobular subtype was about 22.9%. In 212 women (93.3%), we found a tumour ER expression histochemical (H)-score of ≥100 H-score. Of the breast tumours, 71% were progesterone receptor (PR) positive. Overall, 75.7% of women demonstrated HER2 expression of IHC 0/1+, with the remaining 24.3% of women having HER2 expression of IHC 2+, but the *her-2/neu* oncogene was not amplified with fluorescent in situ hybridization (FISH). The CDK4/6 inhibitor of choice included palbociclib (71%), ribociclib (20%) and abemaciclib (9%), whereas partnering aromatase inhibitors were letrozole (63%), anastrazole (28%) and exemestane (9%). The median follow-up duration was 49 months.

### 3.2. Survival Analysis

#### 3.2.1. Whole Cohort

There was no difference in mPFS (*p* = 0.3) (Figure 1A) or mOS (*p* = 0.67) (Figure 1B) in patients who received palbociclib (mPFS = 27.5 months, mOS = 49.5 months) or ribociclib (mPFS = 25.7 months, mOS = 50.2 months). For the 19 patients who received abemaciclib, mPFS or mOS was not reached (Figure 1A,B). The 5-year PFS% was 20.88%, 32.58% and 66.8% for palbociclib, ribocicilib and abemaciclib, respectively (Appendix A). The 5-year OS% was 48.54%, 42.33% and 66.86% for palbociclib, ribocicilib and abemaciclib, respectively (Appendix A).

#### 3.2.2. De Novo versus Recurrent Disease

De novo breast cancers may be biologically distinct and have better survival outcomes compared with recurrent breast cancer [27]. In the whole cohort, mPFS was significantly better in de novo disease compared with recurrent disease (47.1 months vs. 20.3 months, *p* = 0.0002) (Figure 2A). The 5-year PFS% was 34.7% and 15.37% in de novo and recurrent disease, respectively (Appendix A). Similarly, mOS was significantly better in de novo compared with recurrent disease (77.4 months vs. 37.3 months, *p* = 0.0003) (Figure 2B). The 5-year OS% was 65.84% and 36.91% in de novo and recurrent disease, respectively (Appendix A). In patients who received palbociclib, mPFS (43.6 months vs. 20.9 months respectively, *p*-value 0.0017) (Figure 2C), and mOS (77.4 months vs. 36.1 months respectively, *p*-value 0.0034) (Figure 2D) was significantly better in de novo disease compared with recurrent disease. The 5-year PFS% in the palbociclib group was 31.43% and 14.69% in de novo and recurrent disease, respectively (Appendix A). The 5-year OS% in the palbociclib group was 62.38% and 37.98% in de novo and recurrent disease, respectively (Appendix A). In patients who received ribociclib, mPFS (mPFS NR (not reached) vs. 18.85 months, respectively, *p*-value 0.0046) (Figure 2E) and mOS (mOS NR vs. 44.6 months, respectively, *p*-value 0.0378) (Figure 2F) were significantly better in de novo compared with recurrent disease. The 5-year PFS% in the ribociclib group was 74.07% and 19.1% in de novo and recurrent disease, respectively (Appendix A). The 5-year OS% in the ribociclib group was 82.5% and 27.98% in de novo and recurrent disease, respectively (Appendix A). In the abemaciclib group (*n* = 19), mPFS was not reached (Appendix A), and in the de novo cohort, mOS was not reached compared with the recurrence group (Appendix A).

#### 3.2.3. Progesterone Receptor (PR) Status and Survival

ER may regulate the expression of PR in breast cancer cells [28]. We therefore investigated if PR expression status could influence survival outcomes in patients who received CDK4/6i therapy. In the whole cohort, mPFS was significantly better in ER+/PR+ tumours compared with ER+/PR− tumours (38 months compared with 19.2 months, respectively, *p* = 0.0038) (Figure 3A). The 5-year PFS% was 26.73% and 20.27% in PR+ and PR− disease, respectively (Appendix A). Similarly, mOS was significantly longer in ER+/PR+ tumours compared with ER+/PR− tumours (62.6 months vs. 34.3 months, respectively, *p* = 0.02) (Figure 3B). The 5-year OS% was 51.37% and 38.12% in PR+ and PR− disease, respectively (Appendix A). In the subgroup analysis, PR status did not influence survival in patients who received palbociclib (Figure 3C,D). The 5-year PFS% in the palbociclib group was 22.66% and 21.07% in PR+ and PR− disease, respectively (Appendix A). The 5-year OS% in the palbociclib group was 50.41% and 39.48% in PR+ and PR− disease, respectively (Appendix A). In the ribociclib group, ER+/PR− tumours had worse mPFS compared with ER+/PR+ tumours (44 months vs. 10.1 months, respectively, *p* = 0.0014) (Figure 3E) and borderline non-significance for mOS (54.8 months vs. 34.3 months, respectively, *p* = 0.073) (Figure 3F). The 5-year PFS% in the ribociclib group was 41.82% and 13.33% in PR+ and PR− disease, respectively (Appendix A). The 5-year OS% in the ribociclib group was 45.37% and 40% in PR+ and PR− disease, respectively (Appendix A). In the abemaciclib cohort, mPFS (Appendix A) and mOS (Appendix A) were not reached for ER+/PR+ tumours compared with ER+/PR− tumours.

We then investigated PR expression in de novo versus recurrent disease. Interestingly, de novo disease was more likely to be PR+ compared with recurrent disease (*p* = 0.0018) (Table 3). There was no association with ER histochemical scores or HER2 expression (0 or low) (Table 3).

#### 3.2.4. Age and Survival Outcomes

Elderly patients are usually underrepresented in clinical trials [29]. Whether CDK4/6i impact on the survival outcomes in the very elderly (>80 years) is largely unknown. We therefore stratified post-menopausal patients who received CDK4/6i into three distinct age groups: ≤65 years, 66–79 years and ≥80 years. In the whole cohort, there was no significant difference in mPFS (28 months vs. 28 months vs. 21.3 months, respectively, *p* = 0.319) (Figure 4A). The 5-year PFS% was 22.79%, 27.73% and 11.89% in the ≤65 years, 66–79 years and ≥80 years groups, respectively (Appendix A). However, mOS was worse in the ≥80 years group compared with the 66–79 years and ≤65 years groups (35 months vs. 61.7 months vs. 77.4 months, respectively, *p* = 0.016) (Figure 4B). The 5-year OS% was 52%, 51.16% and 32.28% in the ≤65 years, 66–79 years and ≥80 years groups, respectively (Appendix A). In the palbociclib subgroup, we observed that mPFS and mOS was significantly worse in the very elderly (≥80 years) compared with the 66–79 years and ≤65 years groups ((mPFS = 14.5 months vs. 28.2 months vs. 30.2 months, respectively, *p* = 0.01, Figure 4C), (mOS = 29.6 months vs. 61.7 months vs. 77.4 months, respectively, *p* = 0.0002, Figure 4D)). The 5-year PFS% in the palbociclib group was 25.56%, 25.61% and 0% in the ≤65 years, 66–79 years and ≥80 years groups, respectively (Appendix A). The 5-year OS% was 55.27%, 50.17% and 23.34% in the ≤65 years, 66–79 years and ≥80 years groups, respectively (Appendix A). There was no difference in the ribociclib group (Figure 4E,F). The 5-year PFS% in the ribociclib group was 27.93%, 34.29% and 60% in the ≤65 years, 66–79 years and ≥80 years groups, respectively (Appendix A). The 5-year OS% was 36.86%, 49.36% and 60% in the ≤65 years, 66–79 years and ≥80 years groups, respectively (Appendix A). In the abemaciclib cohort, mPFS was not reached (Appendix A) and mOS was not reached for the 66–79 years and ≤65 years groups (Appendix A).

#### 3.2.5. ER Expression Level, Histopathology Types and Survival

The histopathological subtypes of ER+ breast cancer may differentially respond to endocrine therapy [30]. Whether the intensity of expression of ER could also influence the clinical benefit of CDK4/6I is unknown. Therefore, we evaluated histopathology subtypes and ER expression levels in patients who received CDK4/6i. As shown in Appendix A, we did not observe any difference in survival based on ER expression levels (H-scores 0–99 vs. ≥100). We also did not observe any difference between ductal, lobular or mixed subtypes of ER + breast cancers (Appendix A).

#### 3.2.6. Multivariate Analysis of Survival

De novo disease, PR status, age, ER scores, histopathology and CDK4/6i were included in the analysis. As shown in Table 4, de novo disease and age were independently associated with OS. 

## 4. Discussion

CDK4/6 inhibitors have changed the treatment landscape for ER+/HER2− advanced breast cancers. Palbocilib, ribociclib and abemaciclib show similar PFS benefits in clinical trials [17,18,31]. However, similar OS survival benefits have not been demonstrated. For ribociclib [22], a median OS of 63.9 months was demonstrated in a clinical trial [22]. For abemaciclib, a median OS of 67.1 months was shown [23]. For palbociclib, the median OS was only 53.9 months [24]. Differences in trial designs and missing survival data issues (such as loss of follow-up and withdrawal of consent) have been postulated to account for the observed differences in OS. Moreover, distinct biological activities of various CDK4/6 inhibitors may also account for observed differences in OS. Whilst ribociclib and abemaciclib have preferential blockade towards CDK4 compared with CDK6, palbociclib inhibits CDK4 and CDK6 at similar concentrations [32]. However, whether these differences in mechanisms of action can alter breast cancer biology during chronic treatment remains to be determined. Therefore, for a practising oncologist, uncertainty remains as to which CDK4/6 inhibitors to choose in routine clinical practice. In the current study, the overall demographics are comparable between palbociclib and ribociclib groups. We have an older patient cohort (median age 69 years) compared with the MONALEESA-2 [22] and PALOMA-2 [24] studies (median age 62 years). Our long-term follow-up data provide evidence that mPFS and mOS are comparable between palbociclib and ribociclib (mPFS (27.5 months vs. 25.7 months, *p* = 0.3) or mOS (49.5 months vs. 50.2 months, *p* = 0.67)). As fewer number (*n* = 19) of patients received abemaciclib at our centre, we were unable to make reliable mPFS and mOS estimations for this cohort.

The large US-based Flatiron Health database real-world retrospective study evaluated 2888 patients who received either first-line palbociclib plus an AI or AI alone [33]. The mOS was significantly longer in the palbociclib+ AI group compared with AI alone (49.1 months versus 43.2 months, *p* < 0.0001). The mPFS was 19.3 months (17.5–20.7) in the palbociclib+ AI group compared with 13.9 (12.5–15.2) months in the AI alone group in that retrospective study [33]. The Ibrance Real World Insights (IRIS) multinational retrospective study included 1621 patients who received first-line palbociclib plus an AI. The 12-month PFS was 88%, the 12-month survival rate was 96% and the clinical benefit rate (CBR) was 96% for the palbociclib+ AI group. The data compare favourably with data from the PALOMA-2 study [24] (ORR, 42%; CBR, 67%). In the MARIA study (RWD in Italian and German patients), 12-month progression-free rates (PFRs) were 80% in patients treated with first-line P+AI. A recently published study of 206 patients evaluated the clinical benefit of palbociclib (*n* = 96), ribociclib (*n* = 54) and abemaciclib (*n* = 56) after 42 months of follow-up [34]. Whilst abemaciclib was associated with a significant PFS benefit in endocrine-resistant patients and those without visceral involvement, there were no other statistically significant differences among the palbociclib, ribociclib or abemaciclib groups [34]. Taken together, these RWD studies as well as ours provide reassuring evidence that all three CDK4/6i are effective treatment options in this population.

We also made additional important observations in the current RWD study. A significant improvement in survival outcomes was observed in de novo disease compared with recurrent disease irrespective of the CDK4/6 inhibitor therapy used. Although MONALEESA-2 [22] and PALOMA-2 [24] reported PFS benefits in the de novo group, specific mPFS or mOS details were not reported in the main study. We also observed that de novo disease was more likely to be PR+. Interestingly, in our cohort, PR status was also a significant predictor of PFS and OS benefits from CDK4/6 inhibitors plus AI therapy. In the MONALEESA-2 trial [22], PR+ tumours had a mOS of 57.5 months compared with 37.7 months in the PR− group. Our data showed that the mOS in the PR− group was 34.3 months and in PR+ tumours, mOS was 62.6 months. The data suggest that PR expression status could be a simple clinical tool for predicting benefits from AI+CDK 4/6 therapy. The mechanism of sensitivity or resistance to endocrine therapy and CDK4/6 inhibitor therapy is complex [16,35]. ER+/PR− breast cancers respond less well to endocrine therapy compared with ER+/PR+ tumours. Non-functional ER-mediated pathways, HER2− or EGFR-mediated signalling and the loss of PTEN may all contribute to PR negativity and resistance [35]. Resistance to CDK4/6i therapy may also be related to the loss of retinoblastoma (Rb) gene function, increased CDK6 expression, alterations in the cyclinE/CDK2 axis and the hyperactivation of FGFR, RAS, ERBB2, PTEN and AKT1 [16]. Whilst we speculate that PR negativity could be a surrogate marker of aggressive biology and resistance, detailed pre-clinical mechanistic studies will be required to confirm these findings.

There is an underrepresentation of the very elderly population in clinical trials [29]. For example, in the landmark MONALEESA-2 [22] and PALOMA-2 trials [24], less than half of the patients were over 65 years of age. In contrast, 57% of our patients were aged 65 or older and most importantly, 15% (*n* = 35) were very elderly and aged 80 or above. In the very elderly (≥80 years), particularly in the palbociclib subgroup, we observed that mPFS and mOS were significantly shorter compared with the 66–79 years or ≤65 years age groups. The reduced mOS in the very elderly may reflect limited systemic chemotherapy options offered beyond progression or death from other unrelated causes. In the ribociclib cohort, very elderly patients aged 80 or above appear to have better PFS and OS, which is likely related to the lower number of patients in this group. Whilst these data provide interesting insights, larger multicentre real-world studies will be required to confirm our observations. Moreover, age-related factors along with genomic and/or transcriptomic factors may influence survival outcomes in patients who receive CDK4/6i therapy.

Our real-world study has several limitations as well as strengths. It is retrospective and had a lower number of patients on ribocilib or abemaciclib therapy. Nevertheless, our retrospective study provides insights into a real-world PFS and OS comparison between palbociclib and ribociclib. However, a major limitation is that it is not a controlled trial, and caution should be exercised before drawing comparative conclusions. In contrast to multinational medical record collection studies, our single-centre experience does not suffer from selection bias. All patients who received CDK4/6i and AI were included in the analysis. The follow-up of patients in our study was long, thereby enabling reliable mPFS and mOS evaluation.

## 5. Conclusions

In conclusion, the real-world data presented here demonstrates that there is no significant difference in median PFS (27.5 months vs. 25.7 months, *p* = 0.3) or median OS (49.5 months vs. 50.2 months, *p* = 0.67) in patients who received either palbociclib or ribociclib, respectively. De novo disease is significantly associated with prolonged median PFS and median OS compared with recurrence disease (47.1 months vs. 20.3 months (*p* = 0.0002), and 77.4 months vs. 37.3 months (*p* = 0.0003), respectively). PR− tumours have significantly reduced median PFS and OS compared with PR+ disease (19.2 months vs. 38 months (*p* = 0.003), and 34.3 months vs. 62.6 months (*p* = 0.02), respectively). In the very elderly (>80 years), the median PFS and OS are significantly shorter compared with patients who are 65 years or younger (14.5 months vs. 30.2 months (*p* = 0.01), and 77.4 months vs. 29.6 months (*p* = 0.009), respectively) in the palbociclib group. Taken together, our study suggests that all available CDK4/6i remain viable options in ER+/HER2− advanced breast cancers. Whilst de novo breast cancers may have a better clinical outcome and the very elderly may have shorter survival, larger studies are required to confirm our observation.

## Figures and Tables

**Figure 1 cancers-15-05164-f001:**
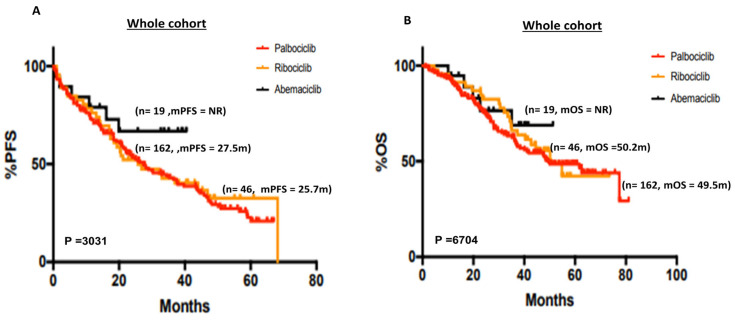
(**A**) Kaplan–Meier curve of progression-free survival in patients who received palbociclib, ribociclib or abemaciclib. (**B**) Kaplan–Meier curve of overall survival in patients who received palbociclib, ribociclib or abemaciclib.

**Figure 2 cancers-15-05164-f002:**
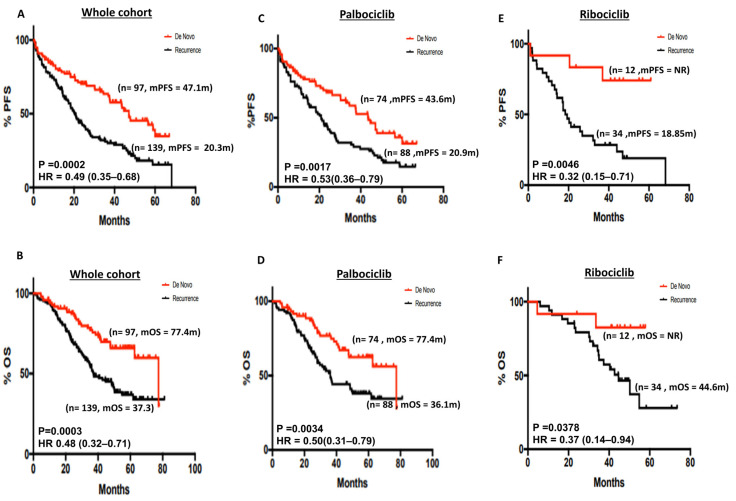
(**A**) Kaplan–Meier curve of progression-free survival in de novo versus recurrent disease in the whole cohort. (**B**) Kaplan–Meier curve of overall survival in de novo versus recurrent disease in the whole cohort. (**C**) Kaplan–Meier curve of progression-free survival in de novo versus recurrent disease in patients who received palbociclib. (**D**) Kaplan–Meier curve of overall survival in de novo versus recurrent disease in patients who received palbociclib. (**E**) Kaplan–Meier curve of progression-free survival in de novo versus recurrent disease in patients who received ribociclib. (**F**) Kaplan–Meier curve of overall survival in de novo versus recurrent disease in patients who received ribociclib.

**Figure 3 cancers-15-05164-f003:**
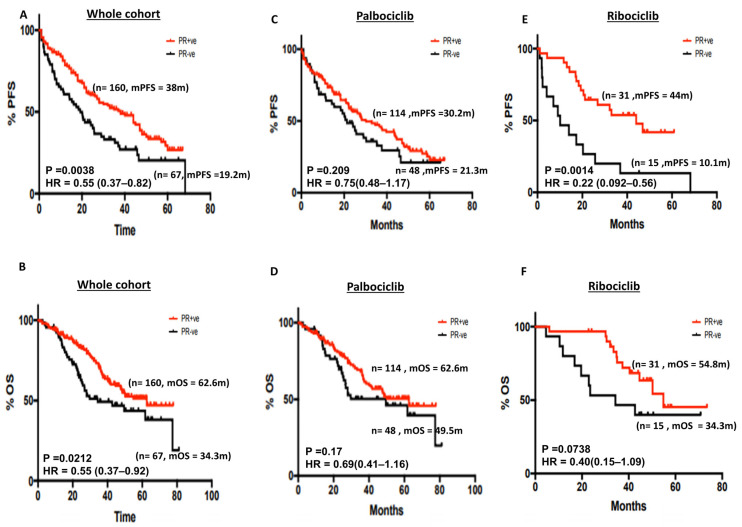
(**A**) Kaplan–Meier curve of progression-free survival in PR+ versus PR+ disease in the whole cohort. (**B**) Kaplan–Meier curve of overall survival in PR+ versus PR+ disease in the whole cohort. (**C**) Kaplan–Meier curve of progression-free in PR+ versus PR+ disease in patients who received palbociclib. (**D**) Kaplan–Meier curve of overall survival in PR+ versus PR+ disease who received palbociclib. (**E**) Kaplan–Meier curve of progression-free survival in PR+ versus PR+ disease in patients who received ribociclib. (**F**) Kaplan–Meier curve of overall survival in PR+ versus PR+ disease in patients who received ribociclib.

**Figure 4 cancers-15-05164-f004:**
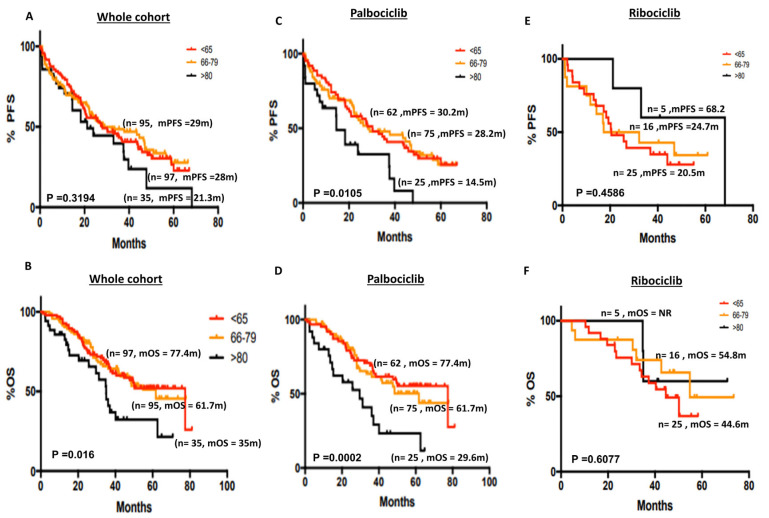
(**A**) Kaplan–Meier curve of progression-free survival based on age (≤65 years versus 66–79 versus ≥80 years) in the whole cohort. (**B**) Kaplan–Meier curve of overall survival based on age (≤65 years versus 66–79 versus ≥80 years) in the whole cohort. (**C**) Kaplan–Meier curve of progression-free survival based on age (≤65 years versus 66–79 versus ≥80 years) in patients who received palbociclib. (**D**) Kaplan–Meier curve of overall survival based on age (≤65 years versus 66–79 versus ≥80 years) in patients who received palbociclib. (**E**) Kaplan–Meier curve of progression-free survival based on age (≤65 years versus 66–79 versus ≥80 years) in patients who received ribociclib. (**F**) Kaplan–Meier curve of overall survival based on age (≤65 years versus 66–79 versus ≥80 years) in patients who received ribociclib.

**Table 1 cancers-15-05164-t001:** Patient demographics.

	Whole Cohort	Palbociclib	Ribociclib	Abemaciclib
	(*n* = 227)	(*n* = 162)	(*n* = 46)	(*n* = 19)
	N (%)	N (%)	N (%)	N (%)
Median age (range)	69 (27–90)	72 (35–90)	64 (27–88)	70 (47–85)
<65 yr	97 (42.7)	62 (38.3)	25 (54.3)	8 (42.1)
65–79	95 (41.9)	75 (46.3)	16 (34.8)	6 (31.5)
80 or more	35 (15.4)	25 (15.4)	5 (10.9)	5 (26.3)
Type of advanced disease				
De novo	97 (42.7)	74 (45.7)	12 (26.1)	11 (57.9)
Recurrence	130 (57.3)	88 (54.3)	34 (73.9)	8 (42.1)
Aromatase Inhibitor used with CDK 4/6 inhibitor therapy				
Letrozole	143 (63.0)	89 (54.9)	38 (82.6)	16 (8.4)
Anastrazole	63 (27.8)	56 (34.6)	5 (10.9)	2 (10.5)
Exemestane	21 (9.2)	17 (10.5)	3 (6.5)	1 (5.3)
Total lines of treatment after progression on CDK4/6i—N (%)				
1	110 (48.4)	75 (46.3)	20 (43.5)	15 (78.9)
2	53 (23.3)	38 (23.4)	11 (23.9)	4 (21)
3	31 (13.6)	26 (16)	5 (10.9)	0 (0)
4	17 (7.5)	13 (8)	4 (8.7)	0 (0)
5	8 (3.5)	4 (2.5)	4 (8.7)	0 (0)
6	6 (2.6)	4 (2.5)	2 (4.3)	0 (0)
7	2 (0.8)	2 (1.2)	0 (0)	0 (0)

**Table 2 cancers-15-05164-t002:** Histopathological characteristics.

Hormone Receptor Status				
ER +ve -N (%)	227 (100)	162 (100)	46 (100)	19 (100)
PR +ve -N (%)	160 (70.5)	114 (70.4)	31 (67.4)	15 (78.9)
PR −ve -N (%)	56 (24.7)	42 (25.9)	11 (23.9)	3 (15.8)
PR unknown -N (%)	11 (4.8)	6 (3.7)	4 (8.7)	1 (5.3)

**Table 3 cancers-15-05164-t003:** ER/PR/HER2 status in de novo and recurrent disease.

	De Novo	Recurrence	Total	Test	Chi-Square
PR +ve	79 (34.8%)	81 (35.68%)	160	Chi-square, df	9.778, 1
PR −ve	18 (7.93%)	49 (21.59%)	67	z	3.127
Total	97	130	227	*p*-value	0.0018
				*p*-value summary	**
				One- or two-sided	Two-sided
				Statistically significant (*p* < 0.05)?	Yes
ER 0–99	4 (1.76%)	7 (3.08%)	11	Chi-square, df	0.1915, 1
ER > 100	93 (40.97%)	123 (54.19%)	216	z	0.4377
Total	97	130	227	*p*-value	0.6616
				*p*-value summary	ns
				One- or two-sided	Two-sided
				Statistically significant (*p* < 0.05)?	No
HER2 negative	36 (15.86%)	36 (15.86%)	72	Chi-square, df	2.277, 1
HER2 low	61 (26.87%)	94 (41.41%)	155	z	1.509
Total	97	130	227	*p*-value	0.1313
				*p*-value summary	ns
				One- or two-sided	Two-sided
				Statistically significant (*p* < 0.05)?	No

‘**’ = *p* < 0.01, ns = not significant.

**Table 4 cancers-15-05164-t004:** Multivariate analysis for overall survival.

	HR	95% CI	*p*-Value
De Novo	0.455	0.2849 to 0.7086	0.0007
PR status	0.6824	0.4508 to 1.047	0.0745
CDK4/6i	0.7747	0.4628 to 1.243	0.3088
Age	1.393	1.044 to 1.848	0.0225
ER score	1.018	0.5214 to 2.533	0.9433
Histopathology	1.081	0.6181 to 1.625	0.8457

## Data Availability

Raw data will be made available upon reasonable request.

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
