# Peer review of "Clinical Impact of CDK4/6 Inhibitors in De Novo or PR− or Very Elderly Post-Menopausal ER+/HER2− Advanced Breast Cancers"

_cancers, 2023, doi:10.3390/cancers15215164_

Round 1
Reviewer 1 Report
This study analyzes the impact of CDK4/6 inhibitors on the progression-free survival (PFS) and overall survival (OS) of patients with advanced breast cancer. The authors reviewed 227 clinical trials and focused on postmenopausal patients, categorizing them based on various factors such as age, ethnicity, ER/PR status, HER2 status, type of disease, and type of aromatase inhibitor used. They used KM curves and a log-rank test to compare the results of patients who received CDK4/6 inhibitors.
Here are some comments:
1. The article did not provide specific data and arithmetic procedures for some of the P-values in the paper.
2. Although the authors stressed the importance of real-world data, they lacked controls for Abemaciclib due to limited data available.
3. The authors only examined the effect of one factor (type of CDK4/6 inhibitor) on PFS and OS, without considering other factors.
4. The introduction was too brief, and the authors must include recent research and compare their findings.
5. The Materials and Methods section should provide more detailed information about the experiments.
6. The Conclusions section needs to present more definitive findings and insights.
7. Some paragraphs lacked proper left-right alignment.
Author Response
We would like to thank the reviewers for very helpful comments. We have addressed all the issues raised by the reviewers and a revised manuscript has been submitted for consideration.
Reviewer 1:
This study analyzes the impact of CDK4/6 inhibitors on the progression-free survival (PFS) and overall survival (OS) of patients with advanced breast cancer. The authors reviewed 227 clinical trials and focused on postmenopausal patients, categorizing them based on various factors such as age, ethnicity, ER/PR status, HER2 status, type of disease, and type of aromatase inhibitor used. They used KM curves and a log-rank test to compare the results of patients who received CDK4/6 inhibitors.
Here are some comments:
- The article did not provide specific data and arithmetic procedures for some of the P-values in the paper.
Response to reviewer: We have clarified the methods and revised (lines 123-125).
- Although the authors stressed the importance of real-world data, they lacked controls for Abemaciclib due to limited data available.
Response to reviewer: We agree with the reviewer. A limitation to our manuscript is we did not have enough patients on abemaciclib to make robust evaluations. However, as requested by the reviewer, we have conducted survival analysis and have presented the data in Supplementary Figure S1. Additional text have been included in the revised manuscript (lines 150-153, 190-192, 239-240, 288-289).
- The authors only examined the effect of one factor (type of CDK4/6 inhibitor) on PFS and OS, without considering other factors.
Response to reviewer: The focus of the retrospective audit was to evaluate the clinical benefit of palbociclib, ribociclib and abemaciclib in post-menopausal patients who received a CDK4/6i. To evaluate clinical benefit, we evaluated progression free survival and overall survival. In addition, we have presented sub-group analysis based on PR status, de novo disease and based on age at presentation. As advised by reviewer 3 we have also completed multivariate analysis (Table 4). De novo disease and age were independently associated with survival.
- The introduction was too brief, and the authors must include recent research and compare their findings.
Response to reviewer: Thank you. We have expanded introduction and included recent research (lines 39-62).
- The Materials and Methods section should provide more detailed information about the experiments.
Response to reviewer: We have revised materials and methods section (lines 78-105).
- The Conclusions section needs to present more definitive findings and insights.
Response to reviewer: We have revised conclusion as advised by the reviewer (lines 420-433).
- Some paragraphs lacked proper left-right alignment.
Response to reviewer: We have corrected this.
Reviewer 2 Report
Manuscript Number: Cancers- 2632998
Title: “Clinical impact of CDK4/6 inhibitors in de novo or PR- or very elderly post-menopausal ER+/HER2- advanced breast cancers”
Even though the objective of the manuscript is within the scope of Cancers, there are minor issues needs to be address in the manuscript before accepting for publication.
Following are the detailed critique of the manuscript
1. Is this study approved by ethics committee? If so, include in the methodology
2. Table representation is very confusing for the readers, authors should separate the Patients demographics and Histopathological demographics in separate tables
3. Authors should expand methodology section. Elaborate methods with different subsections like Patients: elaborates about patients selection and exclusion criteria and location etc., Data collection: How did you collected patients’ data. Histology: what method used for the hormonal status and HER status Statistics analysis.
4. Lane 164-167, is table number representation is correct?
5. Elaborate your results and discussion. Include more latest references throughout the manuscript.
Overall impression
Overall, Methods, Results and discussion are convincing. Conclusion drawn out of the study is satisfactory. This manuscript may accept after minor revision editors’ consideration in the Cancers Journal.
Author Response
We would like to thank the reviewers for very helpful comments. We have addressed all the issues raised by the reviewers and a revised manuscript has been submitted for consideration.
Reviewer 2:
Title: “Clinical impact of CDK4/6 inhibitors in de novo or PR- or very elderly post-menopausal ER+/HER2- advanced breast cancers”
Even though the objective of the manuscript is within the scope of Cancers, there are minor issues needs to be address in the manuscript before accepting for publication.
Following are the detailed critique of the manuscript
- Is this study approved by ethics committee? If so, include in the methodology
Response to reviewer: The retrospective clinical audit study was undertaken at Nottingham University Hospitals with approval from the service evaluation and quality improvement department (audit approval ID: 21-622C). We have revised text in the methods to clarify this.
The audit did not require ethical approval (lines 78-80).
Table representation is very confusing for the readers, authors should separate the Patients demographics and Histopathological demographics in separate tables
Response to reviewer: Thank you, we have separated to Tables 1 (demographics) & 2 (histopathologic characteristics).
- Authors should expand methodology section. Elaborate methods with different subsections like Patients: elaborates about patients selection and exclusion criteria and location etc., Data collection: How did you collected patients’ data. Histology: what method used for the hormonal status and HER status Statistics analysis.
Response to reviewer: Thank you, we have expanded this section as suggested by the reviewer (lines 78-105)
- Lane 164-167, is table number representation is correct?
Response to reviewer: Thank you. We have included the appropriate table in the revision.
- Elaborate your results and discussion. Include more latest references throughout the manuscript.
Response to reviewer: Thank you.
We have updated results as suggested by reviewers 1, 2 and 3 with references were appropriate (lines 150-153, 170-171, 173-174, 176-177, 180-183, 187-192, 220-222, 224-225, 227-228, 230-233, 236-240, 268-270, 273-274, 276-277, 281-289, 320-324, 329-331).
We have revised conclusions (lines 373-386).
Overall impression
Overall, Methods, Results and discussion are convincing. Conclusion drawn out of the study is satisfactory. This manuscript may accept after minor revision editors’ consideration in the Cancers Journal.
Response to reviewer: Thank you.
Reviewer 3 Report
In Table1 Race is not equally divided, some of the groups just have one patient, I would suggest to remove since it doesn't provide any information about RACE.
Similarly remove Her2+.
Most of events in survival plot occurs within 60 months, the authors may modify the data with five years OS/PFS. Since most of the patients are very elderly, we could expect
those patients may die because of age but not due to disease.
fig4 E The authors showed Ribociclib PFS, patients >80 years are doing better when compared to other group, This may be due to less number of patients. In PFS is calculated for first 3 or 5 years, the authors may consier to change this.
Age factor alone will not decide the sensitivity or PFS/OS of the patients, it may be the genomic/transcriptomics alterations, etc.
The present form of manuscript is shows only very basic information, it doesn't provide more information about how the specific drug works to increase/decrease the PFS or OS.
I would recommend the authors to perform in-depth analysis to analyze multi-factors study to prove the hypothesis.
Check carefully
Author Response
We would like to thank the reviewers for very helpful comments. We have addressed all the issues raised by the reviewers and a revised manuscript has been submitted for consideration.
Reviewer 3:
In Table1 Race is not equally divided, some of the groups just have one patient, I would suggest to remove since it doesn't provide any information about RACE.
Response to reviewer: We have removed this from table 1
Similarly remove Her2+.
Response to reviewer: We have removed HER2 status (now able 2 in revised version)
Most of events in survival plot occurs within 60 months, the authors may modify the data with five years OS/PFS. Since most of the patients are very elderly, we could expect those patients may die because of age but not due to disease.
Response to reviewer: As suggested by the reviewer we have included a supplementary table S1 showing 5 years OS% and 5 years PFS%. We agree with the reviewer that the very elderly may die because of age and related co-morbidities rather due to disease (please see 357-359).
fig4 E The authors showed Ribociclib PFS, patients >80 years are doing better when compared to other group, This may be due to less number of patients. In PFS is calculated for first 3 or 5 years, the authors may consider to change this.
Response to reviewer: We agree with the reviewer that the PFS in ribociclib cohort may represents fewer patients. As shown in supplementary table S1, the five-year OS% and PFS% remains at 60% in >80 years. We have included additional text in the discussion section (lines 359-361)
Age factor alone will not decide the sensitivity or PFS/OS of the patients, it may be the genomic/transcriptomics alterations, etc.
Response to reviewer: We agree with the reviewer. We have inserted new text in the discussion section (lines 362-364)
The present form of manuscript is shows only very basic information, it doesn't provide more information about how the specific drug works to increase/decrease the PFS or OS.
Response to reviewer: Thank you. We have inserted additional text in discussion section to highlight potential differences in the mechanism of action of various CDK4/6i (lines 304-311.
I would recommend the authors to perform in-depth analysis to analyze multi-factors study to prove the hypothesis.
Response to reviewer: As suggested by the reviewer we have conducted multivariate analysis. De novo disease, PR status, age, ER scores, histopathology, CDK4/6i were included in the analysis. Denovo disease and age were independently associated with survival. We have included section 3.2.5 and table 4 in results section (lines 298-303)

Round 2
Reviewer 1 Report
The authors have considered all of the comments I provided and made the necessary revisions. Additionally, they have addressed all the concerns.
Reviewer 3 Report
Most of the suggested comments were addressed in the revised version of the manuscript.
Most of the suggested comments were addressed in the revised version of the manuscript. This manuscript may be considered for publication.